# Development and Validation of the Vietnamese Children’s Short Dietary Questionnaire to Evaluate Food Groups Intakes and Dietary Practices among 9–11-Year-Olds Children in Urban Vietnam

**DOI:** 10.3390/nu14193996

**Published:** 2022-09-26

**Authors:** Thi My Thien Mai, Quoc Cuong Tran, Smita Nambiar, Jolieke C. Van der Pols, Danielle Gallegos

**Affiliations:** 1School of Exercise and Nutrition Sciences, Queensland University of Technology (QUT), Victoria Park Rd., Kelvin Grove, QLD 4059, Australia; 2Ho Chi Minh City Center for Disease Control, Ho Chi Minh City 700000, Vietnam; 3Department of Nutrition and Food Safety, Faculty of Public Health, Pham Ngoc Thach Medical University, Ho Chi Minh City 700000, Vietnam; 4Woolworths Centre for Childhood Nutrition Research, Queensland University of Technology (QUT), Graham St., South Brisbane, QLD 4101, Australia

**Keywords:** validation, short dietary questionnaire, food groups, dietary practices, children, Vietnam

## Abstract

This study aims to develop and assess the reproducibility and validity of the Vietnamese Children’s Short Dietary Questionnaire (VCSDQ) in evaluating food groups intakes and dietary practices among school-aged children 9–11 years old in urban Vietnam. A 26-item questionnaire covering frequency intakes of five core food groups, five non-core food groups, five dietary practices over a week, and daily intakes of fruits, vegetables, and water was developed. Children (*n* = 144) from four primary schools in four areas of Ho Chi Minh City, Vietnam completed the VCSDQ twice, as well as three consecutive 24 h recalls over a week. Intra-class correlation, Spearman correlation, weighted kappa, cross-classification, and Bland–Altman plots were used to evaluate the reproducibility and validity. The direct validity of food groups from VCSDQ against the 24 h recalls was examined using Wilcoxon-test for trend. The VCSDQ had good reproducibility in 12 out of 15 group items; the ICC ranged from 0.33 (grains) to 0.84 (eating while watching screens). This VCSDQ had low relative validity, two items (instant noodles, eating while watching screens) had a moderate to good agreement (k = 0.43, k = 0.84). There was good direct validity in three core-food groups (fruits, vegetables, dairy) and three non-core food groups (sweetened beverages, instant noodles, processed meat). In addition, the VCSDQ can also be used to classify daily intakes of fruits and vegetables from low to high.

## 1. Introduction

Childhood overweight and obesity is a global public health issue affecting more than 340 million children aged 5–19 years [1]. The most significant rise in the prevalence of childhood overweight and obesity has been in low-middle income countries [2]. Childhood overweight and obesity is responsible for short and long-term impacts on children’s health, including increased risk of cardiovascular diseases, diabetes, cancer, psychological and mental disorders, and increases the probability of becoming an adult with overweight and obesity [2,3]. In Vietnam, a lower-middle income country in Asia, the prevalence of overweight and obesity in children 5–19 years old has doubled from 8.5% in 2010 to 19% in 2019, with nearly 30% of children living in urban areas being overweight or obese [4]. Notably, in Ho Chi Minh City, the largest city in Vietnam, the prevalence of overweight and obesity in primary school children in 2014 was 51%. Of these, 27% of children were obese, doubling every five years since 2004 [4]. Childhood overweight and obesity is, therefore, a serious public health issue in urban Vietnam. The rise in childhood overweight and obesity has coincided with national economic growth and the co-occurring nutrition transition with rapid changes towards a less healthful diet [5]. This has been primarily the case in urban Vietnam, where this economic growth is characterized by increased intakes of ultra-processed foods and eating outside of the home in emerging multinational fast-food chains [5]. The rapidity of the changes in diet and food environments has been difficult to monitor, making timely public health action challenging.

Promoting a healthy diet is one strategy to prevent the increase in overweight and obesity. However, there is little information on food intake and dietary practices among Vietnamese children. Current dietary assessment in national surveys uses 24 h recalls, which are costly, time-consuming, and tend to focus on nutrients rather than food intake and dietary patterns. In addition, the Vietnamese national nutrition survey is only conducted every ten years and has therefore not been able to keep up to date with emerging trends [6]. These factors limit the opportunity to provide salient data regarding children’s food intake and dietary practices in the context of overweight and obesity that could inform public health policy and strategies. Thus, dietary assessment tools that are low cost, easy to administer and able to provide in-time data about children’s food intakes and dietary practices are needed.

Short dietary questions, used individually or together as a questionnaire, have previously been used as a tool to evaluate and monitor population food intakes and compliance with dietary guidelines [7,8]. Most short dietary questions focus on specific food groups such as fruits and vegetables, dairy groups, or sweetened beverages, as well as some dietary practices such as meal skipping rather than taking a whole diet approach [9]. In addition, most short dietary questions have been developed and validated among children in high-income countries where the food context (environment and intake) differs from those in lower-middle-income countries. Consequently, our primary aim was to develop a short dietary questionnaire to evaluate usual food group intakes and dietary practices over a one-week period for children aged 9–11 years old. The secondary aim was to examine the relative validity and reproducibility of the newly developed short dietary questionnaire against three 24 h food recalls among fifth-grade students in urban Vietnam.

## 2. Materials and Methods

This study follows the Best Practices for Conducting and Interpreting Studies to Validate Self-Report Dietary Assessment Method, which is partly adapted from the checklist for nutritional epidemiology study (STROBE-nut) [10]. See Appendix A for the STROBE-nut checklist.

### 2.1. Study Design

A validation study design was used to examine the reproducibility and relative validity of the Vietnamese Children’s Short Dietary Questionnaire (VCSDQ) among children aged 9–11 years in Ho Chi Minh City. The VCSDQ was validated against three non-consecutive 24 h recalls collected within one week (two weekdays and a weekend). The reproducibility of the VCSDQ was tested using repeated administration one week apart.

### 2.2. Setting

This study occurred at primary schools located in urban and rural areas within Ho Chi Minh City, Vietnam. Ho Chi Minh City is the most populous city in Vietnam, with over nine million people, population density and economic development [11,12]. Due to disruptions caused by the COVID-19 pandemic, data was collected in two periods (July 2020 and September–October 2020).

### 2.3. Participants

A sample size of approximately 160 was required to examine the correlations of intake frequencies between the VCSDQ, and the repeated 24 h recalls (*p* < 0.05, 80% power) [13,14]. A response rate of about 50% was expected based on a previous study [15], consequently, a minimum of 320 students were invited to participate in the research.

Grade-five students were recruited because students in this age group (9–11 years old) have been identified to be able to report their diet using a food frequency questionnaire more accurately than their parents [16]. In Vietnam, primary school starts at Grade 1 when children are 6–7 years old, and secondary school starts at Grade 6 when they are 11–12.

Multi-stage sampling was applied to select participating schools. Firstly, based on geographical categorizations, the city was divided into four areas: four wealthy urban districts, nine less wealthy urban districts, six emerging urban districts, and five rural districts. Four districts were randomly selected from each of these four areas. Then, three primary schools (one for selection and two for backup) per district were randomly selected (*n* = 48). After approaching four selected schools, three agreed to participate in the study. Another school from the list of backup schools in the same district was invited to participate in the study. To achieve an equal sample of student participants from each school *(n* = 40), two to three grade five classes were randomly selected to participate based on the school size.

Information and consent forms were given to 447 children and sent home to the parents/caregivers. Only 163 children who returned two completed consent forms (one from the parent and one from the child) within a week participated in the research. Exclusions included children with cognitive impairments who were unable to complete the survey independently and children with scoliosis or other musculoskeletal disorders or who were restricted to a wheelchair due to the inability to collect accurate anthropometric data.

### 2.4. Development of a Vietnamese Children’s Short Dietary Questionnaire (VCSDQ)

The VCSDQ was developed in English by TMTM, based on the Vietnamese food-based dietary guidelines to cover food groups and dietary practices [17]; the Food Pyramid for children aged 6–11 years old for recommendations of vegetables, fruit serving sizes and water intake [18]; and by reviewing available short dietary questions for children aged 9–11 years old [7,8], internationally to identify common items and structure of short dietary questions. The first draft of the VCSDQ consisted of 29 open- and closed-ended questions on the frequency and serves of food groups intake and mealtime behaviors over the last week. This draft was then translated into Vietnamese and evaluated for content validity by five nutrition experts (local Vietnamese academics with experience in dietary data collection with children). Experts were asked to rate the relevance and clarity of each item using a four-point ordinal scale: 1 = not relevant/clear, 2 = somewhat relevant/clear, 3 = quite relevant/clear, 4 = highly relevant/clear [19]. Three out of five experts rated the VCSDQ as 3, and two rated the questionnaire as 4. The suggestions from experts rated as 3 included improvements in the clarity of languages suitable for the age of primary schools and for Southern dialects as Ho Chi Minh City is in the south of Vietnam. In addition, it was recommended that food items should be re-arranged according to the food groups.

After this process, modifications to the first draft were made, and the second draft of the VCSDQ was reviewed with three children aged 9–11 years old. Cognitive interviewing was used to assess how children understood the questions and their process in answering questions and to identify issues related to understanding the questions and responses. The cognitive interview was recorded, and the main issues were coded using four categories: understanding the question, difficulty related to recalling information, problems with identifying the frequency of consumption or portion size, and difficulty with selecting answer options. Most students found the questionnaire easy to understand but did have difficulty in identifying serves of foods with open-ended questions. Consequently, the revised version of VCSDQ also included three closed questions about the average number of serves of fruits, vegetables and the amount of water consumed daily over the last week, with pictures below each question to illustrate one serve of fruits/vegetables and a cup of water. In addition, to facilitate timely completion and analysis, the paper version of the VCSDQ was designed as an online version using Key Survey™ (Version 8.70, WorldAPP, Quincy, MA, USA) [20]. This version was then translated to English and reviewed by two authors (DG, SNM), and the Vietnamese version was then tested with six students to evaluate readability and the time taken to complete the VCSDQ. All students could answer the VCSDQ by themselves, taking between 6 to 15 min to complete.

The final VCSDQ consisted of 26 items in two languages English and Vietnamese. The first 17 items asked about the frequency of intake of five core food groups (grains, vegetables, fruits, meat and alternatives, dairy) and five non-core food groups (sugar-sweetened beverages, sweets and savory snack foods, fast food, instant noodles, processed meats). The following section included six items about the frequency of five dietary practices (adding salt/sauces, eating a meal with an adult in the family, eating while watching television and other screens, eating a meal cooked outside the home, and skipping meals). The final section was three items about daily intake of fruits, vegetables, and water in serves/cups. Each food group item contained eight response options: not eaten, less than once/week, once/week, 2–4 times/week, 5–6 times/week, one/day, 2–3 times/day, 4 times/day or more. For the six items related to dietary practices, the response options “less than once/week” was removed after cognitive interviewing with children as it was poorly understood. For questions about daily intakes, answer options were added: not eating, less than once serve, once serve/day, 2 serves/day, 3 serves/day, 4 serves/day or more for intake of fruits and vegetables, and options: not drinking water, one cup/day or less, 2–3 cups/day, 4–5 cups/day, 6–7 cups/day, 8 cups/day or more for water intake. In addition, to facilitate the estimation of serves of fruits and vegetables or cups of water, pictures with the example serving sizes of fruit and vegetables and a picture with a cup and bottle of water, were added below the question to increase comprehension and help improve the accuracy of estimation. In addition, to facilitate familiarity with the questionnaires, instructions for answering the questions and two practice questions were added prior to the actual VCSDQ questions. See Appendix A for a copy of the VCSDQ.

### 2.5. Reference Method: 24 h Recalls over Three Separate Days

Children were asked to recall what they ate and drank 24 h before the interview, using the 5-step multiple-pass method developed by the USDA [14]. Each child was interviewed at school three times in one week to collect 24 h recalls from two weekdays and one weekend day. To improve the estimation of the foods and drinks consumed, a book with food pictures showing portion sizes and everyday utensils to help understand portion sizes were used. In addition, to improve the quality of the 24 h recalls, the school menu with the actual picture of one serving and the ingredients were collected before the interview to support interviewers and facilitate children in reporting and estimating their meals consumed at school. All data from the 24 h recalls were recorded on a prepared 24 h data collection sheet by interviewers trained in 24 h recall methods by the lead investigator. Due to the limited time allocation from school and the availability of interviewers during each data collection period, 20 interviewers who were students from the Nutrition and Dietetics and Preventive Medicine programs at the Ho Chi Minh City University of Medicine and Pharmacy were recruited to participate in a two-day training workshop for data collection. After this workshop, each interviewer was asked to submit an actual record of a 24 h recall to be reviewed and provided with suggestions for improvements until the record accurately represented the previous day’s intake. In each period, 8–12 interviewers were invited to collect the 24 h recalls. In addition, two health officers who were experienced with 24 h recall interviews checked for any unusual food intakes or missed food records.

Each 24 h recall record was converted into grams of food intake or mL of beverage consumption using a food weight conversion table. In addition, a book of pictures of common foods for dietary assessment in children developed by the National Institute of Nutrition [21], a book of street foods with usual portion size in Ho Chi Minh City developed by Ho Chi Minh City Nutrition Centre [22], and unpublished conversion databases from Ho Chi Minh City Nutrition Center [23] were used for the estimates. After the first round of data entry, the conversion database of ten food items was unavailable, so the researchers weighed two samples of the food item, and average values were used to determine its weight.

Intake amounts of all reported foods and drinks were entered into a Microsoft Excel Spreadsheet and assigned a food code using the 2017 Vietnamese food composition tables [24]. If food-items were not available in the Vietnamese food composition tables for example, chia seeds or oats, a new food-code was created and the nutrient composition was borrowed from either the ASEAN Food Composition Table, USDA food composition data. In addition, food composition data from countries closest to Vietnam such as Singapore, Thailand, and India or the country from where the food was exported to Vietnam such as Australia or Japan were used for reference. If the food was not available in other food composition tables, then the data was inputted using nutrients reported on the food label if available or was calculated based on the food composition of ingredients for composite foods (e.g., dried chicken with lemon leaf, octopus’ ball, Oreo™ cookies and milk blended with ice).

Each record included: study ID, date of visit, name of the dish, time of eating, name of a meal (breakfast/lunch/dinner/suppers/snack), place of eating (school/home/shop/on the way to school or from school to home), home-cooked (yes/no), food code, name of food, the weight of each food, amount of nutrients, start and end time of the interview, duration of the interview, whether a child was consuming their usual diet or not, whether the child was following a special diet or not. In addition, information about whether children ate meals (breakfast, lunch, dinner) with family members or watched screens at these meals during the 24 h recall period was recorded to facilitate the validation of the VCSDQ.

### 2.6. Matching Food Groups and Dietary Practices from the 24 h Recall with the VCSDQ

To facilitate the comparison between the VCSDQ and the 24 h recalls, all data from the 24 h recalls, and the VCSDQ were converted into frequencies per day for the 17 items of frequency intakes and the six of dietary practices items.

For the VCSDQ, the frequencies per day was converted using the conversion table (Table 1):

For the 24 h recalls, the frequencies per day were calculated using the following steps.

Firstly, the frequency intakes of each food group for each 24 h recall were calculated based on the number of eating occasions. Each eating occasion was defined as all food and drink consumed within a 30 min interval [26]. For example, if vegetables were consumed on two eating occasions across 24 h, the number of vegetable intakes over 24 h were counted as twice. However, with two different types of vegetables consumed on one eating occasion, the number of vegetable intakes was still counted as once [26]. All primary ingredients from the 24 h recalls were allocated into one of the 17 food items (five core and five non-core food groups) as reported in the VCSDQ by the lead author (TMTM) and reviewed by other authors (DG, SMN, JVDP) using the food classifications from the Vietnamese Food Composition Table [24]. The number of food groups intakes over 24 h was calculated by the total number of intakes of those food groups for each eating occasion.

For dietary practices, the number of times the child ate with other family members and watched screens while eating during breakfast, lunch or dinner were calculated from the same question in the 24 h recalls. Other dietary practices, including adding salt, adding sauces while eating, eating outside the home, and skipping meals, were defined, and coded based on the information from the 24 h recalls. Adding salt was defined as the number of eating occasions that the child added salt while eating fruit (it is a cultural practice to eat fruits with salt instead of using table salt to add to foods in western countries) or adding fish or soya sauce while eating other dishes. Adding sauces was defined as the number of eating occasions that children added other sauces such as tomato sauces, chilli sauces, and mayonnaise while eating. Eating outside of the home was defined as the number of eating occasions that cooked foods were consumed at shops or restaurants or home but were not homemade. Skipping meals was defined as the number of times children did not eat anything for breakfast, lunch or dinner.

Secondly, as the VCSDQ asked about the frequency of intakes/eating habits over the last week, the daily number/weight of intakes/dietary practices from the 24 h recalls were adjusted for a whole week using the following equation: average of intakes from two weekdays multiplied by five plus intakes on the weekend multiplied by two and this total of a week’s intake was divided by seven. Ten children had only two 24 h recalls; a third record was created using the average value of the other two records prior to applying this equation. If a child had only one 24 h recall, they were excluded, but there were no cases with one record. In addition, the daily intake of fruits, vegetables, and water from 24 h recalls was calculated to facilitate the comparison with three items from the SDQ (mean daily intake of fruits, vegetables, and water over the last week).

### 2.7. Anthropometric Measurements

All participating children had weight, height, and waist circumference measured twice by trained health officers using standard measurement protocols [27]. If the difference between two measurements for height was 0.5 cm, for weight 0.5 kg, and waist circumference 1 cm or more, a third measurement was conducted. Height was measured by using a wooden height board to the nearest 0.1 cm, weight was measured using scales (TANITA HD-318, TANITA, Tokyo, Japan) to the nearest 0.1 kg, waist circumference was measured by non-plastic tapes to the nearest 0.1 cm, at the end of normal expiration at the midpoint between the lower margin of the last palpable rib and the top of the iliac crest.

#### Covariates

Child characteristics: sex and date of birth were recorded during the anthropometric assessment.

Overweight and obesity: data from the anthropometric measurements were used to generate BMI z-scores for age using the WHO Anthropometric macro [28]. According to recommendations from WHO [28], overweight was defined as BMI z-for-age > 1 SD, and obesity as BMI z-for-age > 2 SD.

### 2.8. Procedures

Data collection took place in a private location at the school suitable for interviewing and the collection of anthropometric measurements over three visits (with one visit on Monday to collect weekend diet data). The first visit and the third visit were one week apart. The 24 h recalls were collected in three visits by the interviewer with children’s reports. The VCSDQ was completed by children at the first and third visit with the support from research team members using Key Survey™ App, Version 8.7, WorldAPP, Quincy, MA, USA on tablets. The process of validation study is presented in Figure 1.

### 2.9. Bias

The food intake records from the average of three 24 h recalls were reviewed to determine under- and over-reporting using Goldberg’s method [29,30]. The under- and over-reporting of dietary intake were identified by comparing the ratio of reported energy intake (EIrep) to the predicted basal metabolic rate (BMRest) with the 95% upper and lower confidence limit of physical activity level (PAL):$$\mathrm{PAL}\times \exp\;[SDmin\times \frac{S/100}{\surd n}\;<\;\mathrm{EIrep}:\;\mathrm{BMRest}\;<\;\mathrm{PAL}\times \exp\;\left[SDmax\times \frac{S/100}{\surd n}\right]$$

The mean of energy intake from the three 24-h recalls for each student was used as estimated energy intake. The predicted basal metabolic rate was calculated for each child using established equations adjusted for weight, height, relevant sex, and age groups in consideration the context of overweight and obesity in the study population [31,32]. The 95% lower and upper limit were calculated with the physical activity level was 1.55 due to the evidence of the low level of physical activity among fifth-grade children in Ho Chi Minh City [15] and the average within-subject variation in intake (S) calculated by the equation below, with suggested values for within-subject variation in energy intake (*CV_wEI_*), within-subject variation in repeated BMR measurement (*CV_wB_*), total between-subject variation in PAL (*CV_tP_*), and the number of days of dietary assessments were 23%, 8.5% and 15%, 3 days, respectively [33].
$$S=\sqrt{\frac{CV_{wEI}^{2}}{d}+CV_{wB}^{2}+CV_{tP}^{2}}$$

### 2.10. Statistical Analyses

The total frequencies of intake per day for the core and non-core food groups, and the frequency per day of the different assessed dietary practices, were used for the analysis. The distribution of all variables resulting from the VCSDQ and the 24 h recalls were examined. If variables did not have a normal distribution, the median of frequencies rather than the mean was presented. Consumption frequencies per day from VCSDQ and the 24 h-recalls were converted to quartiles for subsequent analysis.

#### 2.10.1. Reproducibility

The average intra-class correlation (ICC) (two-way mixed-effect model, absolute agreement) was used to examine the reproducibility of the VCSDQ after one week. ICC values were categorized as <0.5, 0.5 to <0.75, 0.75 to <0.9, and ≥0.9, indicating that the questionnaire had poor, moderate, good, or excellent reproducibility for the food group analyzed, respectively [34]. In addition, Spearman correlation coefficients, classification of agreement, and weighted kappa were applied to further examine the reproducibility of the VCSDQ.

#### 2.10.2. Relative Validity

The relative validity of VCSDQ was examined by comparing intake frequencies with those derived from the average of the three 24 h recalls as the reference method. Group-level validation tests including Wilcoxon paired signed-rank test, and individual–level validation tests, including Spearman correlation, classification agreement, and weighted kappa to examine the relative validity of the VCSDQ [35,36].

For group-level validation, the equality of the median between the first VCSDQ and 24 h-recalls was examined using Wilcoxon paired signed-rank test as all data were skewed (*p* > 0.05 indicating values from two methods are equal) [35,37]. In addition, Bland-Altman plots were used to evaluate the direction of the bias (under and over-reporting of the VCSDQ compared to three 24 h recalls) at group level by estimating mean differences (VCSDQ minus 24 h recalls) in times per day with 95% limit of agreement (mean ± 1.96 SD) [35,38].

For individual-level validation, the correlation between VCSDQ and 24 h recalls were firstly examined by Spearman rank correlation coefficient (SCC) to evaluate the strength and direction of the association (>0.5 good outcomes, 0.2–0.49 acceptable outcomes, and <0.2 poor outcomes) [35,36]. Then, to adjust the within-person variation from 24 h-recalls, de-attenuation correlations were calculated using the equation [14]:rt=r01+λxnx
where *r_t_* is the de-attenuation correlation, *r*_0_ is the observe correlation, *λ_x_* is the ratio of the within- and between-person variances for the 24 h recalls *n_x_* is the number of repeated measurements of 24 h recalls (*n_x_* = 3).

Cross-classification was used to test whether the child was correctly classified in the same quartile (exact agreement), the same and ±1 quartile (exact and adjacent agreement) or the opposite quartile (gross misclassification). Good agreement was indicated when the percentage was in the same quartile > 50% or opposite quartile < 10% [35,36]. To further examine the classification agreement, quadratic weighted kappa values were calculated. The strength of the agreement, as defined by Landis and Koch [39], was categorized as value ≤ 0 (no agreement), 0.01–0.20 (slight agreement), 0.21–0.40 (fair agreement), 0.41–0.60 (moderate agreement), 0.61–0.80 (substantial agreement), 0.81–0.99 (perfect agreement).

#### 2.10.3. Direct Validity

Although examining responsiveness to change is generally encouraged in the validation of diet questionnaires, it was not possible to assess change over longer periods in this cross-sectional study. Instead, the direct validity with sensitivity to change was estimated by analyzing mean intakes calculated from 24 h recall across categories of the VCSDQ and examining the trend by applying the Wilcoxon-test for trend, which is an extension of the Wilcoxon rank-sum test [40]. We used this method to assess the existence of any trends in intake of fruits, vegetables, and water from 24 h-recalls across increasing number of serves of intake from the VCSDQ.

All analyses were carried out in STATA statistical software version 17 (StataCorp LLC, College Station, TX, USA). Statistical significance was defined as *p*-value < 0.05.

### 2.11. Ethics

The study was conducted under the Declaration of Helsinki and approved by the Queensland University of Technology Human Research Ethics Committee (protocol version 3, approved 30 September 2019; UHREC Reference number: 1900000601).

## 3. Results

### 3.1. Participants

One hundred and sixty-three (*n* = 163) fifth-grade students from four schools in Ho Chi Minh City participated in the study, giving a response rate of 36.5%. Of these, 153 (93.4%) participants completed three 24 h recalls, with the remaining participants completing two recalls. Nineteen (11.6%) participants who misreported (4.3% under-reported and 7.3% over-reported) were excluded from the analysis. After excluding mis-reporters, 144 students were included in the analysis of relative validity. For the analysis of VCSDQ’s reproducibility, five students were absent for the second administration of the VCSDQ, so a total of 139 participants were included in the analysis. A summary of participant numbers for each study is presented in Figure 2.

The characteristics of participants and mis-reporters are presented in Table 2. Boys and girls were evenly represented, and the mean of age was 10.6 ± 0.5 years. Nearly sixty percent (58.9%) of participants were overweight or obese with 31% obese. Median energy intake was 1916 kcal/day, which is higher than in the under-reporter groups (1150 kcal/day) and lower than over-reporters groups (3121 kcal/day). The proportion of children having school lunch among participants was higher than those in the under and over-reporters group. However, there were no significant differences between included participants and those who misreported their dietary intake.

### 3.2. Descriptive Data

Overall, the frequencies of intakes and dietary practices, and daily food intakes between the VCSDQ and 24 h recalls were significantly different across most food groups and dietary practices except for the frequency intakes of fruits, instant noodles, and skipping meals (Table 3). Records from 24 h-recalls indicated that children consumed fruits, dairy products, sweetened beverages, snacks, and discretionary foods, had family meals, and ate out of home at least once per day. They consumed vegetables twice a day and grains, meat, and alternatives three times a day. In contrast, frequency intakes of these food groups and dietary practices from the VCSDQ were all less than once a day. Conversely, the daily intakes of fruits, vegetables, and water estimated from the VCSDQ were significantly higher than from the 24 h recalls.

### 3.3. Reproducibility

The VCSDQ had moderate to good reproducibility with an ICC of 12 out of 15 items >0.5 (Table 4). The frequency of watching television or other screens while eating had the highest reproducibility value with an ICC = 0.84 (0.78–0.89). Three items with poor reproducibility were intake frequencies for grains, processed meats, and the frequency of eating outside of the home (ICC < 0.5).

At an individual level, all items except for eating grains had an acceptable Spearman correlation between the first and second administration (SCC ≥ 0.3). Only two items (intake frequency of fruits and frequency of watching television or other screens while eating) had a good correlation (SCC ≥ 0.5). In addition, 12 out of 15 items (80%) had a gross misclassification lower than 10%. Further examination of the agreement between the first and second administration of the VCSDQ, the weighted kappa showed that all items had at least fair agreement, one out of four core food groups, three out of five non-core food groups, and four out of five dietary practices had a moderate to a substantial agreement without a chance.

### 3.4. Relative Validity

Overall, the VCSDQ had a low relative validity in most food group intakes and dietary practices. The de-attenuation Spearman correlation ranged from −0.06 for frequency intake of grains to 0.73 for frequency of watching while eating. Seven out of 15 items had an acceptable association (de-SCC > 0.5) (Table 5). By using weighted kappa, only two items (frequency intake of instant noodles and frequency of watching while eating) had a moderate to good relative validity (weighted kappa = 0.43 (0.29–0.57), 0.69 (0.59–0.8), respectively). Although having a fair agreement (0.2 < weighted kappa < 0.4) in the frequency intake of vegetables, dairy, sweetened beverages, processed meat (weighed kappa = 0.23, 0.3, 0.29, and 0.21, respectively), the percentages of gross misclassification among these food groups were less than 10% (4%, 7%, 5% and 5%, respectively). The percentage of exact and adjacent agreement ranged from 58–76% in core-food groups, 68–76% in non-core food groups, and 54–90% in dietary practices. Six items with a high percentage of gross misclassification (≥10%) were grains, sweets and savory snacks, adding salt/sauces, having a meal with family, eating outside of the home, and skipping a meal.

At the group-level, only median of frequency intake of fruits and the frequency of skipping meals were equal (Table 3). In addition, the examination of agreement in the frequency of intakes and dietary practices between the VCSDQ and 24 h recalls using ICC indicated that only three items (intake frequency of sweetened beverages, instant noodles, and frequency of watching television and other screens while eating) had a moderate to a good agreement (Table 5).

The Bland–Altman plots with mean differences in times/day (VCSDQ minus 24 h recalls) and 95% limit of agreements are presented in Figure 3. The VCSDQ underreported frequency intake of eight out of ten food groups and three out of five dietary practices, and overreported frequency intake of fruits, instant noodles, and skipping meals. The mean differences were large for frequency intake of grains (−1.82 times/day) and meat and alternatives (−1.78 times/day) and were small for frequency of skipping meals (0.03 times/day); instant noodles (0.11 times/day) and processed meat (−0.11 times/day). Five items (frequency intake of fast food, instant noodles, processed meat; and frequency of watching while eating and skipping meal) had a 95% limit of agreement within 2 times/day.

### 3.5. Direct Validity

The ability to detect the trend in frequency intakes of core and non-core food groups is presented in Table 6. Overall, there was an increase in the amount of food intake (g/mL per day) from 24 h recalls between the first quartile and fourth quartile in six out of 10 food groups (vegetables, fruits, dairy, sweetened beverages, instant noodles, processed meat). For example, the amount of intake of sweetened beverages increased by each quartile from the first quartile (178 mL/day) to the second quartile (257 mL/day) to the third quartile (264 mL/day) and to fourth quartile (333 mL/day) indicating a positive trend between the amount intake of sweetened beverages from 24 h-recalls and the frequency intake from the VCSDQ.

The direct validity was also examined with the daily intake of fruits, vegetables, and water. Table 7 shows an increase in the daily amount of intake of fruits and vegetables from the 24 h-recalls corresponding to the increase in serves of fruits and vegetable intakes from the VCSDQ (*p* < 0.05). However, the weight of daily intakes from the 24 h recalls for each response category did not precisely match the servings’ illustrations from the VCSDQ. For example, the average intakes of fruits and vegetables from the VCSDQ in the category 4 serves/day or more were defined as 400 g/day or more, whereas these figures from the 24 h recalls were 82.4 g and 145 g, respectively. Although there was a lower water intake in the category “not drinking water” compared to other categories, this item did not show trend (*p* = 0.163). Noticeably, the amount of water intake in categories “>8 cups/day” from the 24 h recall was about 686 mL/day, which is far lower than the estimated water intake for this category (1600 mL/day).

A summary table of results is presented in Table 8 to facilitate the utilization of each questionnaire item.

## 4. Discussion

This study examines the reproducibility and relative validity of a 26-item Vietnamese Children’s Short Dietary Questionnaire (VCSDQ), which was developed to rapidly evaluate the frequency intakes of five core and five non-core food groups and five dietary practices among children 9–11 years in urban Vietnam over a one-week period.

### 4.1. Reproducibility

Overall, the VCSDQ showed moderate to good agreement for repeated measurement at a group level (ICC > 0.5) for most of the items (except for grains, processed meat and eating outside of the home) and fair to good agreement for repeated measurement at the individual level (weighted kappa > 0.2). These findings are comparable with another short dietary questionnaire in children and adolescents [9]. The low reproducibility of VCSDQ for processed meat (ICC = 0.43) was similar to the results reported in a questionnaire developed for children 9–10 years old in New Zealand (NZ) (ICC = 0.38 [26]. For the intake of grains, there were no other published studies that we could compare our results with. The most similar study was conducted in NZ children showed that the reproducibility of intake estimates for rice and rice-based dishes was high (ICC = 0.73), whereas in our study, the reproducibility for intakes of grains including rice, rice-based dishes, bread, starchy foods was low (ICC = 0.33). This difference in observations between those two countries may be due to the differences in children’s diets between NZ and Vietnam. In NZ rice and rice-based dishes are potentially requested dishes eaten in response to children’s preferences so they may to more likely to be remembered [41]. Conversely in Vietnam, rice and rice-based dishes are staple foods and consumed regularly in the diet. Using the weighted kappa, the reproducibility of cereal/grain intakes (k = 0.2) in our study was lower than that observed in older children (11–12 year old and 13–14 year old) in Belgium (k = 0.55 and 0.58) [42] and (12–17 year old) in China (ICC = 0.48) [43]. This higher reproducibility with older age groups could imply that the cognitive capacity of children aged 9–10 may not be fully adequate to quantify the frequency of grain/cereal intake. In addition, the fact that grains (rice, rice noodles, rice paper rolls, etc.) are usually include a range of food items (which may be difficult for young children to discern), are often eaten in different eating contexts and in mixed dishes [9]. This makes the quantification of this food group more difficult, particularly for children. Low reproducibility of estimating cereal intake was also reported in pregnant women (ICC = 0.25) and lactating women in China (ICC = 0.48) [44,45]. Thus, it is possible that estimating the intake of grains and cereals has low reproducibility in Asian countries.

For the reproducibility of dietary practices, four of five items had good reproducibility except for eating outside of the home (ICC = 0.47, k = 0.3). Presently there are no other studies to compare these results with directly. For a similar concept of eating “take away foods”, the reproducibility was low in Australian Aboriginal children (k = 0.39) but higher in non-Aboriginal children (k = 0.59) [40]. In China, eating outside of the home was positively associated with overweight and obesity in children (6–17 years old) [40], and with dietary energy from fat and high sugar intake in Vietnamese adolescents (15–17 years old) [46]. However, the questions for eating outside of the home had not been validated or had been drawn from 24 h recalls [46,47]. Thus, apart from examining food intake, questionnaires to examine dietary practices potentially highly relevant to children’s nutritional status and health should be validated to enable high-quality dietary data collection in the context of the nutrition transition in low- and middle-income countries.

### 4.2. Relative Validity

Overall, the relative validity of this first version of the VCSDQ was generally low, with eight out 15 items having a poor agreement, five having a fair agreement, and two having a moderate to good agreement at the individual level. Three out of 15 items had a good agreement at the group level. Similarly, low validity of short dietary questionnaire items was found in other studies [9,26].

For the agreement at the individual level, two core food groups (frequency intakes of vegetables, dairy) and three non-core food groups (sweetened beverages, snacks, processed meat) had a fair agreement (k = 0.21–0.4), one non-core food group (instant noodles) had a moderate agreement (k = 0.43), and one dietary practice (watching screens while eating) had a good agreement (k = 0.69). Although frequency intakes of four non-core food groups and frequency of watching screens while eating were lower than frequency intakes of core-food groups, more items in non-core food groups have a higher validity than core-food groups. This comparison could indicate that these items are children’s food preferences, so they are more likely to report them accurately than core-food groups [48]. In the context of high prevalence of overweight and obesity in children, the potential preference for non-core foods as well as the inaccuracy in estimating core-food groups are both issues of concern impacting on reporting. Although children’s nutritional status can influence children’s reporting of dietary intake [41], this analysis has not yet been completed for this study. Further analysis needs to be conducted to examine factors associated with the accuracy of child report of core and non-core foods groups in the context of high prevalence of overweight and obesity.

Eight out of ten food groups (vegetables, fruits, meat and alternatives, dairy, sweetened beverages, fast-foods, instant noodles, processed meat) and one out of five dietary practices (watching screens while eating) had gross misclassification of less than 10%. All items were able to allocate more than 50% of individuals in the same or adjacent group, which indicated acceptable agreement between the two methods at the individual level. So, although having a fair agreement (weighted kappa = 0.21–0.4), above nine items with misclassification less than 10% or exact/adjacent agreement more than 50% could be fairly used to classify children’s intakes from the VCSDQ in a similar group (exact or one quartile difference) from the 24 h recalls. Compared with other studies, our study had a lower gross misclassification than two studies from New Zealand and Belgium [26] and had similar results to a study in China [43]. Although the Spearman correlation coefficient is not a standard statistic for evaluating the agreement between two methods, we used this to compare with other similar studies. Our study had comparable results for most food groups, including fruits, vegetables, meats and alternatives, sweetened beverages, snacks, and adding sauces but lower values for processed meat, rice and rice-based dishes, dairy, and higher values for instant noodles compared to a study in New Zealand [26].

For agreement at the group level, three items (frequency of intake of fruits, instant noodles, and eating outside of home) had good agreement, whereas the median intake of most food groups and dietary practices from the VCSDQ were significantly lower than from the 24 h recalls. The Bland–Altman plots also indicated that the VCSDQ underreported the frequency of food group intakes and dietary practices compared to 24 h recalls.

The explanation for the good agreement of these items could again be due to children’s food preferences and the regular consumption of fruits and instant noodles [41,48]. Although the frequency of fruits and vegetables intakes was not high, these food items may be eaten on a regular basis at similar times, for example fruits were often eaten after meals as desert and instant noodles were often eaten in the break time at school or at breakfast/supper. In addition, if children have a specific preference for fruits and instant noodles (either liking or disliking), it is potentially easier to recall the frequency intakes of these food items [41]. Reporting fruit intake is also more accurate among children who are overweight or obese [49], so this could be one of explanation in our study sample given that about 60% of participants were either overweight or obesity. In Ho Chi Minh City eating outside of the home is common dietary practice, which is associated with special events in the family or associated with children’s requests restaurants/shops due to the foods or location or environments of eating [50]. So, this dietary practice is potentially more likely to be accurately reported by children.

For the underestimation of frequency intakes from the VCSDQ, one of potential explanation is the retention interval [41]. Children may be more accurately remember what they ate during 24 h recall than over one-week recall from the VCSDQ. In addition, the 24 h recall was administrated by interviewer with the prompt and aids to recall, whereas the VCSDQ was self-administered by children. As a result, most of food items or dietary practices would be omitted in the VCSDQ. In addition, children also had a lower capacity to remember food items when eating out of the home [51], so the food items associated with dietary practices may be also omitted in the VCSDQ. Another possible explanation could also be due to the high prevalence of overweight and obesity as those children are likely to underreport their diet [52].

Although the food frequency questionnaire tends to overestimate children’s intake, the under or overreporting of frequency intake of food group and dietary practices from the short dietary questions has not been fully investigated. However, one study in Australian children (4–11 years) reported by parents found that the amount of food group intakes (fruits, vegetables, bread and cereals, meat and alternatives, dairy and extra foods) and diet index score from short food questions were significantly higher than from 24 h recalls [53]. The overreporting of daily amount of intake was similar our study where the daily intake of fruits, vegetables, and water from the VCSDQ were significantly higher than from the 24 h recalls. However, in our study, the frequency intake of food groups from VCSDQ was significantly higher than from 24 h recalls and the underreport tends to be larger in main food groups including grains, meat and alternatives. So, future analysis should examine factors associated with misreporting from short dietary questions.

The frequency of grains, meat and alternatives intakes had a low validity at both the group and individual levels. In other studies, the validity of meat and grain intakes was also low. The Pearson correlation coefficients between a short food questionnaire and three 24 h recalls in Australian children (4–11 years old) assessed as servings/day were 0.08 and 0.07, respectively [53]. The same comparisons in Chinese children (12–17 years old) were 0.13 (cereals), 0.37 (red meat), −0.04 (poultry), 0.14 (seafoods) [43]. A possible explanation for the varying levels of agreement for each questionnaire item is the number of items belonging to a food group and the culture of shared meals in Asian countries. Items with many sub-groups and a wider variety of foods such as grains or meats and meat alternatives seem to have lower validity, whereas items that are clearly defined such as instant noodles had a higher validity. In addition, grains (particularly rice and rice-based dishes) and meat and alternatives (main courses) were often eaten in a shared meal with other family members. So, children may find it more difficult to recall accurately which types of foods and how much of these foods they consumed.

Consequently, to improve the validity of the VCSDQ, food groups such as grains and meat and alternatives, a strategy could be to divide them into discrete sub-groups that more clearly indicate the most obvious and commonly consumed food items within that group using a standard recipe and serving. Additional cognitive interviewing is also recommended to understand children’s perceptions of the classification of foods and their portion sizes. In addition, if the data is available, a food list should ideally be developed from the actual dietary intakes of children to provide examples of common dishes and food items. Such a food list should preferably be compiled from data collected in the sub-group of the total population of interest before developing the short dietary questions for dietary assessment [54,55]. These questionnaire modifications could improve the validity of items currently showing low validity in the VCSDQ.

Validity of frequency estimates of dietary practices such as adding salt/sauces, eating with family members, and skipping meals is low. So far, these dietary practices have not been items of interest in other validation studies. In the New Zealand study discussed above, the questionnaire item “frequency intakes of tomato sauces, ketchup” had a low validity similar to our sauces-use question (SCC = −0.11) [26]. In Vietnam, these sauces are often eaten with fried foods (fried meatballs, fried chicken, chips) or fast foods (pizza, burgers). Apart from tomato sauces, fish sauces or soy sauces are often presented on the table for dipping with foods (fried foods or steamed foods) or adding to a dish (pho, rice noodles) as a typical eating habit in Vietnamese meals. In the validation of dietary practices, the ‘watching screens while eating’ item had high validity, whereas other dietary practices associated with mealtimes had low validity. So, it is likely that if children are not conscious about their diet or are watching screens while eating, they may not be conscious of whether they add sauces to their food or whom they eat with, thereby influencing their answers in the VCSDQ. In addition, eating with family and skipping meals are two behaviors associated with mealtimes, so it is possible that the perception of mealtimes among children in this age group is not clearly defined. In the context of dynamic family activities and work patterns in urban areas, the structure of mealtimes may not be clear and inconsistent day-by-day, so this may also make it more difficult for children to define mealtime practices. This lack of consistency in the diet can also lead to lower accuracy in estimating food intakes and dietary practices [48].

### 4.3. Direct Validity

Although there are some limitations in the relative validity, the VCSDQ did have the ability to detect trends in the frequency of intakes of dairy, sweetened beverages, instant noodles, and processed meats from low to high quartiles, and for both frequency of intakes and mean daily intakes of fruits and vegetables. These results are comparable to a study among children in Australia (10–12 years old), where mean food intakes from 24 h recalls increased with increasing intake frequencies of pasta/rice, fruit, milk, cheese, butter, red meat, eggs, fruit juice, soft drink, salty snacks, confectionery, and breakfast cereal from the short food frequency questionnaire [40]. So, these items could potentially be used to discriminate the food intake levels of children between low and high in the context of the nutrition transition, where monitoring trends in children’s food intakes over time is essential. Such information is needed to develop timely policies and interventions for promoting healthy core food group consumption (fruits, vegetables, dairy), and reduce the intake of unhealthy food groups (e.g., sweetened beverages, instant noodles, and processed meat).

### 4.4. Strengths and Limitations

#### 4.4.1. Strengths

To date, this is the first study in Vietnam and one of few studies in low- and middle-income countries to develop and validate short dietary questions in evaluating children’s whole diet, including food group intakes and dietary practices in school-aged children aged 9–11 years old.

This study included a sample size comparable to other validation studies (*n* = 144), and this sample was representative of fifth-grade students in Ho Chi Minh City in terms of age, sex, and nutritional status [4,15].

Similar to other validation studies of short dietary questions, this study has used multiple analyses (ICC, SCC, weighed kappa, cross-classification, Bland–Altman plots, Wilcoxon test for trend) to examine the reproducibility and validity of the short dietary questions, thereby providing a comprehensive assessment of this tool.

#### 4.4.2. Limitations

This study’s data collection method highly depended on children’s recall for both the reference method (24 h recall) and the test method (short dietary questionnaire). For the 24 h recall, children were interviewed by trained research assistants using a standard protocol commonly used for 24 h recall interviewing, and the record of 24 h recall was checked by health officers experienced with dietary data collection however, any recall bias due to children’s capacity to remember and retrieve their diet and dietary habits could not be eliminated.

Another limitation of this study is the cognitive capacity of children 9–11 years old to report their own diet. Although the accuracy to recall children’s diet increase by age [41], generally, the ability to report diet among children under twelve-year-old is potentially limited, particularly in estimating portion size [56]. In addition, although children 8–11 years old could report their own diet using food frequency questionnaire better than their parents [16], they still have limited capacity to report their diet particularly if they ate meals outside of the home [51].

Evidence indicates that the use of photos of foods in dietary intake tools can improve the capacity to correctly estimate food portion size when children report their usual diet [57]. We were able to use a number of photographs of typical food and portion sizes, but due to the continuously increasing diversity of food items available, pictures of food portion sizes were not available for all foods to help students correctly estimate the amount of food typically consumed. The available Vietnamese food composition table lacked many of these newly available food items (such as breakfast cereal, hamburger, pizza, meatballs, etc.), so the classification of these food items into food groups from 24 h recall may mismatch with children’s classification of food from the VCSDQ. Such food classification errors are common in low-middle income countries where food composition databases are often insufficient [58].

The cognitive interviewing in this study was conducted with three children who lived in urban areas which may not allow comparison to the perceptions of children living in rural districts where accessibility to fast-food chain restaurants or food items from stores is limited. Therefore, children may have overlooked and misreported their intakes and dietary practices, despite the initial training using example questions. Future development of dietary tools for use in children from low- and middle-income countries should consider cognitive interviewing in a larger sample of children to understand more fully perceptions of food group classifications and portion size estimations as a process of the validation study.

The typical Vietnamese diet includes many mixed dishes and composite foods, so children may find it challenging to classify food groups contained in these mixed dishes, which may also have led to some of the misreporting in this study. A more dish-based assessment could be considered when revising the VCSDQ items as this approach is increasingly used in Asian countries where mixed-dishes are popular [59].

Due to the COVID-19 pandemic, data collection was constrained to a short period of time, and this may have created some level of pressure on the children for self-completion of VCSDQ. In addition, during this time, all interviewers wore masks to prevent COVID-19 transmission, so the ability to build trust and have a comprehensive conversation with children was limited compared to normal circumstances. These conditions may have influenced data collection in this study though this is estimated to be minimal.

Finally, the second VCSDQ was administrated one week after the first VCSDQ, so children may remember what they reported from the first administration. This may lead to the potential increase in the reproducibility of the VCSDQ. However, the increase in interval would reduce the feasibility of the study due to the limited time of data collection and reduce the reproducibility of the VCSDQ due to the change in children’s diet week by week.

## 5. Conclusions

This VCSDQ is one of the first short tools developed in Vietnam. Elements of this tool could be used to evaluate food group intakes and dietary practices in children 9–11 years old. The 26-item VCSDQ had an acceptable reproducibility for all food groups and dietary practices. At the group level, the VCSDQ could be used to rank the frequency intakes of fruits, vegetables, dairy, sweetened beverages, instant noodles, and processed meat. At an individual level, this tool had a fair to good capacity to evaluate frequency intakes of vegetables, dairy sweetened beverages, snacks, instant noodles, processed meat (and frequency of watching screens while eating). Revisions to the VCSDQ need to be made in order to use it as a questionnaire to evaluate dietary intakes and dietary practices among children. These revisions will provide real time data for the development and evaluation of policies and interventions in low- and middle-income countries.

## Figures and Tables

**Figure 1 nutrients-14-03996-f001:**
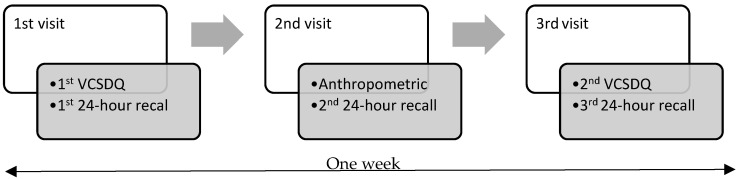
Process of validation study of Vietnamese Children’s Dietary Short Questionnaire.

**Figure 2 nutrients-14-03996-f002:**
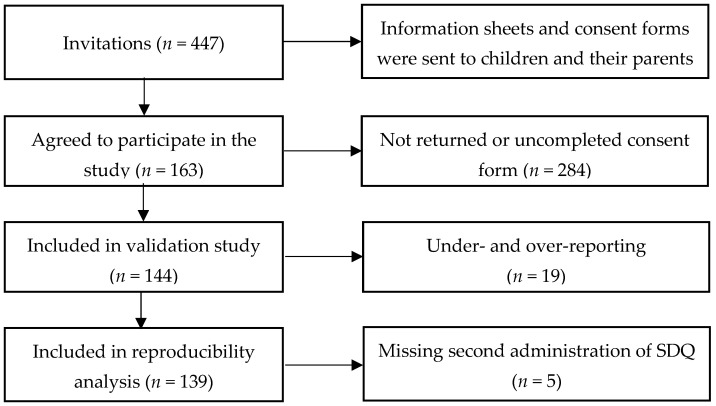
Participants in the validation study.

**Figure 3 nutrients-14-03996-f003:**
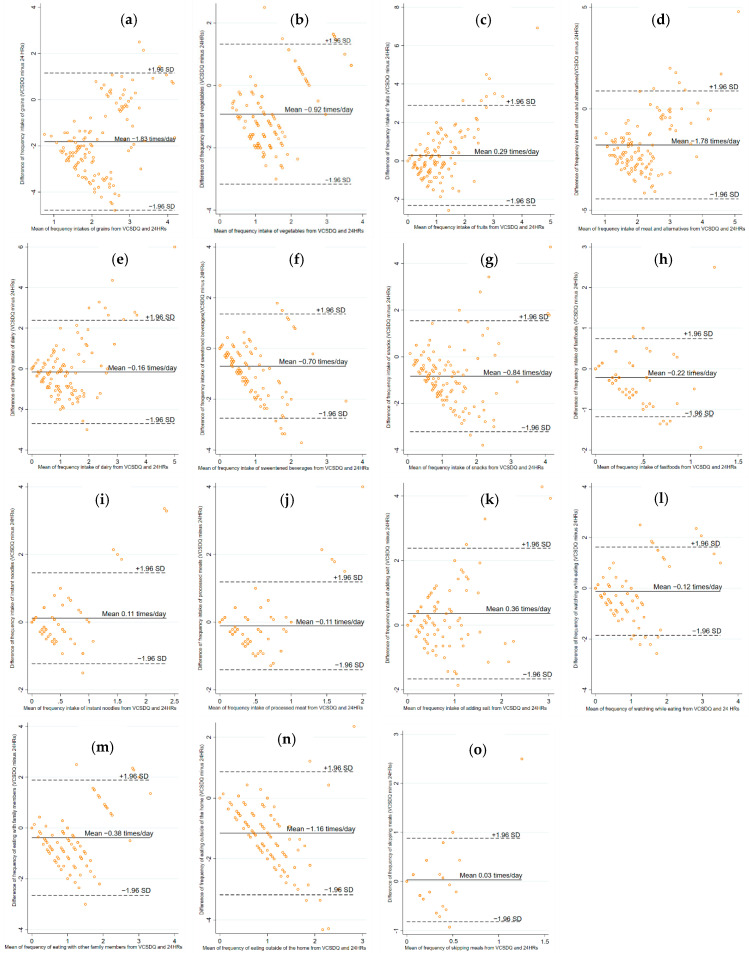
Bland–Altman plots describing the mean differences (VCSDQ minus 24 h recalls) for frequency intake of food groups and dietary practices in times per day: (**a**) grains; (**b**) vegetables; (**c**) fruits; (**d**) meat and alternatives; (**e**) dairy; (**f**) sweetened beverages; (**g**) snacks and discretionary foods; (**h**) fastfoods; (**i**) instant noodles; (**j**) processed meats; (**k**) adding salt; (**l**) watching screens while eating; (**m**) eating with family members; (**n**) eating outside of the home; (**o**) skipping meals. The solid line represents the mean, and the dashed lines represent the 95% limit of agreement (+1.96 SD and −1.96 SD) of the observation. The *y*-axis shows the VCSDQ (test method) minus 24 h recalls (reference method) in times/day, the *x*-axis shows the mean between VCSDQ and 24 h recalls in times/day.

**Table 1 nutrients-14-03996-t001:** Conversion table compilated based on the calculation from [25].

VCSDQ Response	Frequency	Calculation	Times/Day
1	Never/not eating	0	0.00
2	less than once/week	0.5/7	0.07
3	once/week	1.0/7	0.14
4	2–4 times/week	3.0/7	0.43
5	5–6 times/week	5.5/7	0.79
6	once/day	1.0/1	1.00
7	2–3 times/day	2.5/1	2.50
8	≥4 times/day	4/1	4.00

**Table 2 nutrients-14-03996-t002:** Characteristics of participants in the validation of VCSDQ *.

Characteristic	Total Participants (*n* = 163)	Under and Over Reporters (*n* = 19)	Participants in Validation Study (*n* = 144)
Sex, male	81 (49.7)	10 (52.6)	71 (49.3)
Age (year)	10.6 ± 0.5	10.8 ± 0.6	10.6 ± 0.5
Weight (kg)	41.5 (34.1–49.3)	41.5 (33.7–52.7)	41.5 (34.1–48.5)
Height (cm)	143.2 ± 7.1	143.3 ± 7.2	142.9 ± 6.9
BMI z-score (SD)	1.25 (0.2–2.2)	1.3 (0.3–2.2)	1.3 (0.2–1.3)
Nutritional status			
Thinness	5 (3.1)	1 (5.3)	4 (2.8)
Normal	62 (38.0)	8 (42.1)	54 (37.5)
Overweight	46 (28.2)	3 (15.8)	43 (29.9)
Obesity	50 (30.7)	7 (36.8)	43 (29.9)
Energy intake (kcal/day)	1932 (1625–2232)	2581 (1185–3285)	1916 (1645–2178)
Having school lunch	116 (71.2)	10 (52.6)	106 (73.6)

* The data is presented as mean ± SD or median (25th–75th) or *n* (%).

**Table 3 nutrients-14-03996-t003:** Median frequency intake and frequency of dietary practices per day from the 1st VCSDQ and 24 h recalls (*n* = 144).

Item	1st VCSDQ			24 h Recalls × 3 Days			*p*-Value *
	Median	25th	75th	Median	25th	75th	
**Core food groups** (times/day)							
Eating grains	0.64	0.28	2.5	3.00	2.64	3.61	<0.001
Eating vegetables	0.61	0.14	1.00	2.00	1.46	2.36	<0.001
Fruits	0.86	0.43	1.69	1.00	0.64	1.36	0.198
Meat and alternatives	0.83	0.43	1.76	3.00	2.61	3.48	<0.001
Dairy	0.79	0.14	1.22	1.07	0.64	1.71	0.002
**Non-core food groups** (times/day)							
Drinking sweetened beverages	0.14	0.07	0.43	1.00	0.54	1.71	<0.001
Snacks	0.28	0.14	0.86	1.43	1.00	2.00	<0.001
Eating fast food	0.07	0.00	0.14	0.36	0.00	0.64	<0.001
Eating instant noodles	0.14	0.07	0.43	0.00	0.00	0.5	0.183
Eating processed meat	0.14	0.00	0.43	0.36	0.29	0.71	<0.001
**Dietary practices**							
Adding salt	0.57	0.28	1.14	0.29	0.00	0.71	<0.001
Watching screens while eating	0.14	0.00	0.79	0.36	0.00	1.07	<0.001
Eating with other family members	0.79	0.14	1.00	1.29	1.00	1.82	<0.001
Eating outside of the home	0.14	0.14	0.43	1.36	1.00	2.00	<0.001
Skipping meals	0.00	0.00	0.14	0.00	0.00	0.29	1.000
**Daily intakes**							
Fruits (serves/day)	2.00	1.00	2.00	0.73	0.36	1.17	<0.001
Vegetables (serves/day)	2.00	1.00	3.00	0.98	0.69	1.37	<0.001
Water (cups/day)	4.50	2.50	6.50	2.84	1.96	3.71	<0.001

***** Wilcoxon matched-pairs signed-rank test *p* < 0.05 significant, one serve = 100 g, one cup = 200 mL.

**Table 4 nutrients-14-03996-t004:** Reproducibility of the VCSDQ (*n* = 139).

Item	SCC (95%CI)	ICC (95%CI)	Weighted κ (95%CI)	Exact Agreement (%)	Exact and Adjacent Agreement (%)	GM (%)
**Core food groups**						
Eating grains	0.20 (0.02–0.38)	0.33 (0.06–0.52)	0.20 (0.02–0.37)	38	73	11
Eating vegetables	0.36 (0.20–0.52)	0.52 (0.33–0.66)	0.35 (0.19–0.51)	43	77	8
Fruits	0.50 (0.37–0.64)	0.67 (0.53–0.76)	0.50 (0.37–0.63)	43	82	3
Meat and alternatives	0.35 (0.21–0.50)	0.52 (0.33–0.66)	0.35 (0.20–0.51)	37	76	6
Dairy	0.34 (0.18–0.50)	0.51(0.31–0.65)	0.34 (0.18–0.5)	37	78	6
**Non-core food groups**						
Drinking sweetened beverages	0.36 (0.21–0.51)	0.52 (0.33–0.66)	0.35 (0.20–0.50)	42	72	5
Snacks	0.39 (0.24–0.55)	0.57 (0.40–0.69)	0.39 (0.24–0.55)	39	81	9
Eating fast food	0.44 (0.29–0.58)	0.61 (0.46–0.72)	0.44 (0.29–0.59)	45	80	4
Eating instant noodles	0.42 (0.28–0.57)	0.59 (0.43–0.71)	0.42 (0.27–0.57)	40	83	5
Eating processed meat	0.30 (0.14–0.47)	0.43 (0.21–0.60)	0.27 (0.11–0.44)	35	76	7
**Dietary practices**						
Adding salt	0.45 (0.31–0.60)	0.63 (0.48–0.73)	0.45 (0.31–0.60)	41	81	4
Watching screens while eating	0.72 (0.63–0.81)	0.84 (0.78–0.89)	0.73 (0.64–0.82)	62	87	1
Having family meal	0.49 (0.34–0.64)	0.64 (0.49–0.74)	0.46 (0.31–0.62)	55	83	6
Eating outside of the home	0.31 (0.15–0.47)	0.47 (0.26–0.62)	0.30 (0.13–0.46)	53	65	14
Skipping meal	0.46 (0.29–0.63)	0.63 (0.48–0.74)	0.45 (0.28–0.63)	72	77	12

SCC: Spearman correlation coefficient, ICC: intra-class correlation coefficient, GM: Gross misclassification.

**Table 5 nutrients-14-03996-t005:** Relative validity of the VCSDQ against 24 h recalls (*n* = 144).

Item	SCC	De-SCC *	ICC	ICC-Adjusted *	Weighted κ (95%CI)	EG (%)	EAG (%)	GM (%)
**Core food groups**								
Eating grains	−0.06 (−0.23–0.10)	−0.06	−0.19 (−0.65–0.14)	−0.20	−0.09 (−0.25–0.07)	23	58	15
Eating vegetables	0.23 (0.08–0.38)	0.25	0.38 (0.14–0.55)	0.41	0.23 (0.08–0.38)	31	70	4
Fruits	0.17 (0.02–0.33)	0.19	0.30 (0.02–0.49)	0.33	0.17 (0.02–0.33)	26	71	9
Meat and alternatives	0.16 (−1.50–0.32)	0.18	0.28 (−0.01–0.48)	0.31	0.16 (0.00–0.31)	26	65	7
Dairy	0.31 (0.15–0.46)	0.33	0.46 (0.26–0.61)	0.49	0.30 (0.15–0.46)	33	76	7
**Non-core food groups**								
Drinking sweetened beverages	0.30 (0.15–0.45)	0.34	0.45 (0.24–0.60)	0.52	0.29 (0.14–0.43)	33	68	5
Snacks	0.24 (0.08–0.40)	0.25	0.39 (0.15–0.56)	0.41	0.24 (0.09–0.40)	31	68	10
Eating fast food	0.13 (−0.02–0.29)	0.14	0.25 (−0.04–0.46)	0.27	0.14 (−0.01–0.30)	24	69	9
Eating instant noodles	0.44 (0.30–0.57)	0.49	0.60 (0.44–0.71)	0.67	0.43 (0.29–0.57)	43	76	6
Eating processed meat	0.20 (0.06–0.35)	0.22	0.35 (0.10–0.53)	0.38	0.21 (0.05–0.37)	33	75	5
**Dietary practices**								
Adding salt	0.09 (−0.08–0.25)	0.10	0.18 (−0.14–0.41)	0.20	0.10 (−0.07–0.27)	29	67	13
Watching screens while eating	0.71 (0.61–0.80)	0.73	0.82 (0.75–0.87)	0.84	0.69 (0.59–0.80)	56	90	3
Having family meal	0.17 (0.00–0.33)	0.18	0.25 (−0.05–0.46)	0.27	0.14 (−0.02–0.30)	31	65	12
Eating outside of the home	0.03 (−0.13–0.20)	0.03	0.06 (−0.30–0.33)	0.06	0.03 (−0.13–0.19)	32	54	19
Skipping meal	0.06 (−0.11–0.23)	0.07	0.12 (−0.22–0.37)	0.14	0.08 (−0.09–0.26)	56	60	23

* The correlation was adjusted by within-variation of 3 days 24 h recall; SCC: Spearman rank correlation coefficient; ICC: intra-class correlation; De-SCC: de-attenuation; EC: exact agreement; EAG: exact and adjacent agreement; GM: gross misclassification.

**Table 6 nutrients-14-03996-t006:** Daily food group intakes from 24 h-recalls and by quartiles from the 1st VCSDQ.

Item	Weight (g/Day)			Mean Intakes (g/Day) from 24 h Recall by Quartiles of the 1st SDQ				*p*-Value *
	Mean	25th	75th	Q1	Q2	Q3	Q4	
**Core food groups**								
Eating grains	212.17	157.75	254.33	213.71	220.50	210.61	206.10	0.650
Eating vegetables	98.15	68.80	136.61	88.19	93.14	115.83	138.31	<0.001
Fruits	73.45	36.01	117.90	69.02	81.39	90.70	115.80	<0.001
Meat and alternatives	215.42	169.98	268.67	216.38	223.97	225.40	231.16	0.920
Dairy	166.07	78.57	269.46	107.76	192.21	203.44	246.11	<0.001
**Non-core food groups**								
Drinking sweetened beverages	196.20	72.50	368.75	177.98	256.86	263.55	332.99	<0.001
Snacks	67.23	31.96	97.53	60.75	68.59	77.05	76.22	0.160
Eating fast food	22.26	0.00	45.52	24.99	39.28	40.23	34.77	0.070
Eating instant noodles	0.00	0.00	30.38	8.41	17.41	25.71	29.29	<0.001
Eating processed meat	7.20	0.12	14.05	5.94	8.76	9.29	11.93	0.020

* Wilcoxon test for trend using command “nptrend” from Stata17.

**Table 7 nutrients-14-03996-t007:** The correlation between daily intakes of fruits, vegetables, and water from the 24 h recalls with the number of serves of fruits/vegetables and cups of water from the VCSDQ (*n* = 144).

Responses from VCSDQ			1	2	3	4	5	6
			Not Eating	Less Than 1 Serve	1 Serve	2 Serves	3 Serves	≥4 Serves
**Daily intakes from 24 h-recalls × 3 days**	**Fruits**	*n*	7	15	43	46	23	10
		Mean (g/day)	29.5	69.7	87.7	98.6	103.1	82.4
		*p*-value *	0.032					
	**Vegetables**	*n*	10	19	37	36	20	22
		Mean (g/day)	61.1	84.2	99.1	115.4	115.1	145.0
		*p*-value *	<0.001					
	**Water**		**Not Drinking**	**≤1 Cup**	**2–3 Cups**	**4–5 Cups**	**6–7 Cups**	**≥8 Cups**
		*n*	2	19	40	40	24	15
		Mean (mL/day)	276.8	569.5	584.7	550.3	615.5	686.6
		*p*-value *	0.163					

* Wilcoxon test for trend using command “nptrend” from Stata 17.

**Table 8 nutrients-14-03996-t008:** Reproducibility, relative validity and sensitive to trend of each VCSDQ items.

	Reproducibility					Validity					Sensitivity to Trend	Usefulness
	Group Level	Individual Level				Group Level	Individual Level					
	ICC	SCC	Cross-Classification		Weighted Kappa	Signed Test	De-SCC	Cross-Classification		Weighted Kappa		
			GM < 10%	EG > 50%				GM < 10%	EG > 50%			
**Core food groups**												
Eating grains		*			*							NA
Eating vegetables	*	*	*		*		*	*		*	*	RVS
Fruits	*	**	*		**	*		*			*	RVS
Meat and alternatives	*	*	*		*			*				R
Dairy	*	*	*		*		*	*		*	*	RVS
**Non-core food groups**												
Drinking sweetened beverages	*	*	*		*		*	*		*	*	RVS
Snacks	*	*	*		*		*			*		RV
Eating fast food	*	*	*		**			*				R
Eating instant noodles	*	*	*		**	*	*	*		**	*	RVS
Eating processed meat		*	*		*		*	*		*	*	VS
**Dietary practices**												
Adding salt	*	*	*		**							R
Watching screens while eating	**	**	*	*	***		**	*		***		RV
Eating with other family members	**	*	*	*	**							R
Eating outside of the home		*		*	*	*			*			V
Skipping meals	*	*		*	**							R
**Daily intakes**												
Fruits (serves/day)											*	S
Vegetables (serves/day)											*	S
Water (cups/day)												NA

* ICC 0.50–0.75 (moderate intra-class correlation), SCC or de-SCC 0.20–0.49 (acceptable outcomes), cross-classification (good agreement), weighted kappa 0.21–0.4 (fair agreement), signed-test (*p* > 0.05). ** ICC 0.75–0.90 (good intra-class correlation), SCC or de-SCC > 0.5 (good outcome), weighted kappa 0.41–0.60 (moderate agreement). *** weighted kappa 0.61–0.80 (substantial agreement). NA: non-acceptable for use, R: acceptable reproducibility at group level, V: acceptable validity at group level or individual level, S: acceptable sensitiveness to trend.

## Data Availability

The datasets generated and analyzed during the current study is available from the corresponding author on reasonable request.

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
