# Peer review of "Development and Validation of the Vietnamese Children’s Short Dietary Questionnaire to Evaluate Food Groups Intakes and Dietary Practices among 9–11-Year-Olds Children in Urban Vietnam"

_nutrients, 2022, doi:10.3390/nu14193996_

Round 1

Reviewer 1 Report

The validation study of a FFQ follows the conventional method. The analyses were comprehensive and presented clearly. However, it would be desirable to have Bland-Altman plot presented.

1.       The reliability and validity of the FFQ can be inflated as the study was conducted in one week. Children may remember what they reported previously. This should be discussed as a limitation.

2.       The discussion and the conclusion are too long and should be shortened.

3.       Table 2: the significant difference in the median intake assessed by 1 VCSDQ and 24-hour recallsx3 should be discussed. For some items, the difference is ten times. For example, eating outside the home (0.14 vs 1.36).

4.       It is desirable to have a  figure showing the process of the validation study.

Author Response

Dear reviewer,

Thank you for your valuable comments on our manuscript. Please see our response below and refer to the attachment to follow the revision of the manuscript:

Point 1: The validation study of a FFQ follows the conventional method. The analyses were comprehensive and presented clearly. However, it would be desirable to have Bland-Altman plot presented.

Respond to point 1: Thank you for your advice regarding the Bland-Altman plot. The manuscript would be more comprehensive with this analysis. So, we have added the Bland – Altman plots (Figure 2) to the manuscript. Accordingly, method, result and discussion regarding Bland- Altman was also added in lines 353-357 (method), lines 488-199 (result), and lines 627-631 and 646-668 (discussion)

Point 2: The reliability and validity of the FFQ can be inflated as the study was conducted in one week. Children may remember what they reported previously. This should be discussed as a limitation.

Respond to point 2: Thank you for your suggestion. Yes, the time frame of one week may affect the reproducibility of the VCSDQ because children may remember what they reported in the first administration. However, for the validity of the VCSDQ, the time -frame for three days 24-hour recalls should be in one week to be able to match with the time -frame from VCSDQ which asks about children’s diet over a week period. Discussion regarding the limitation of the time frame was presented in lines 778-793.

Point 3: The discussion and the conclusion are too long and should be shortened.

Respond to point 3: Thank you for your suggestion. Yes, we have shortened the conclusion and some areas of the discussion as much as possible.

Point 4: Table 2: the significant difference in the median intake assessed by 1 VCSDQ and 24-hour recallsx3 should be discussed. For some items, the difference is ten times. For example, eating outside the home (0.14 vs 1.36).

Respond to point 4: Thank you for your advice. We have added the discussion on the differences between median intake from VCSDQ and from 24-hour recalls in the relative validity section in conjunction with the Bland-Altman plots (lines 646-668)

Point 5: It is desirable to have a figure showing the process of the validation study.

Respond to point 5: Thank you for your suggestion. We have added a figure to show the process of the validation study in the method section (lines 306-310)

Reviewer 2 Report

Authors present a detailed description of the development and validation of a short questionnaire for use with children 9-11 years of age. The manuscript is organized, methods are transparent and reproducible, and results are thoroughly discussed.  As well, conclusions and possible applications are appropriate. Two minor comments/suggestions:

Besides recall bias, another potential limitation is the ability of children 9-11 years of age to provide an accurate 24-hour recall. Is there any research about this? In some studies, parents or other proxies are present when children this age are interviewed. I’m not sure if this would have substantially changed results though. 

Lines 69-75: Do authors mean short questions or questionnaires

Line 88:  Spell out VCSDQ at first mention.

Lines 325-329: This is a little unclear. Were these the only groups merged? Some appear to be core groups, some non-core groups and some dietary practices. 

Author Response

Dear reviewer,

Thank you for your valuable feedback and encouragement. Please see the responses from us in the text below and please see the attachment to follow the revision of the manuscript:

Point 1: Besides recall bias, another potential limitation is the ability of children 9-11 years of age to provide an accurate 24-hour recall. Is there any research about this? In some studies, parents or other proxies are present when children this age are interviewed. I’m not sure if this would have substantially changed results though. 

Respond to point 1: Thank you for your suggestion. There is existing research regarding the capacity of children 9-11 years old in reporting their own diet. Their ability to accurately recall dietary information is an ongoing debate as results are mixed [1,2]. We had added this in the discussion section, please refer to lines 750-756.

Point 2: Lines 69-75: Do authors mean short questions or questionnaires

Respond to point 2: Thank you for pointing this out. We meant short questions, as short questions could be used individually to evaluate specific food intakes or dietary practices. Alternatively, all questions could be presented as one questionnaire to evaluate the whole diet or to examine adherence to dietary guidelines. We had updated the clarification of short dietary questions in line 70 with the addition of the following: Short dietary questions, used individually or together as a questionnaire, ….

Point 3: Line 88:  Spell out VCSDQ at first mention.

Respond to point 3: We have revised the attached revised manuscript (line 90).

Point 4: Lines 325-329: This is a little unclear. Were these the only groups merged? Some appear to be core groups, some non-core groups and some dietary practices. 

Respond to point 4: Thank you for your question. The VCSDQ include 17 item questions for five core food groups and five non-core food groups and 6 items about the frequency of five dietary practices. This description has been mentioned in the previous section (lines 166 - 172), so we removed the description of merging food items for core, non-core food groups or dietary practices in this section.